# A Systematic Review of the Health and Healthcare Inequalities for People with Intersex Variance

**DOI:** 10.3390/ijerph17186533

**Published:** 2020-09-08

**Authors:** Laetitia Zeeman, Kay Aranda

**Affiliations:** 1School of Health Sciences, University of Brighton, Brighton BN1 9PH, UK; K.F.Aranda@brighton.ac.uk; 2Centre for Transforming Sexuality and Gender, University of Brighton, Brighton BN2 0JG, UK

**Keywords:** intersex, health inequalities, healthcare, LGBTI, ethical accountability, sex, gender

## Abstract

Extensive research documents the health inequalities LGBTI people experience, however far less is known for people with intersex variation. This paper presents a review of intersex health and healthcare inequalities by evaluating research published from 2012 to 2019. In total 9181 citations were identified with 74 records screened of which 16 were included. A synthesis of results spans nine quantitative, five qualitative and two narrative reviews. Literature was searched in Medline, Web of Science, Cochrane, PsycInfo and CINAHL. People with intersex variance experience a higher incidence of anxiety, depression and psychological distress compared to the general population linked to stigma and discrimination. Progressive healthcare treatment, including support to question normative binaries of sex and gender, aids understand of somatic intersex variance and non-binary gender identity, especially when invasive treatment options are avoided or delayed until individuals are able to self-identify or provide consent to treatment. Findings support rethinking sex and gender to reflect greater diversity within a more nuanced sex-gender spectrum, although gaps in research remain around the general health profile and the healthcare experiences of people with intersex variance. More large-scale research is needed, co-produced with peers who have lived experience of intersex variation to ensure policy, education and healthcare advances with greater inclusivity and ethical accountability.

## 1. Introduction

The twenty first century is marked by an increasing awareness of gender, sex and sexual plurality with recognition of how these factors may impact on health and wellbeing. Recent global public health research reflects significant health inequalities for lesbian, gay, bisexual, trans and intersex (LGBTI) people compared to the general population. Large scale international reviews collating these research findings include studies of LGBTI health inequalities [1,2], the disease burden in gender and sexual minorities [3], intersex health [4], the global health burden and needs of transgender populations [5], the health profile of sexual minority women [6], and more recently, loneliness and social support in older LGB communities [7]. These studies share a common denominator of an increasing concern over how diverse identities and bodies intersect and experience health inequalities in LGBTI populations. Inequalities are compounded when LGBTI people access healthcare, with discrimination, heteronormativity or minority stress based on their gender identity, sexual orientation or sex characteristics [8]. Practitioners working in health and social care settings are tasked with understanding the specific needs of LGBTI people in a context of evolving practice, changing terminology and emerging policy directives [9].

Whilst previous research has indicated that LGBTI people experience significant health inequalities impacting both on their health outcomes and on their experiences of healthcare systems [10,11,12], populations within LGBTI health inequalities research constitute a range of non-homogenous groups. For these groups a key priority identified in research is the health of people with intersex variance. Although people with intersex variations are often included in the LGBTI umbrella, community-based organisations and service users actively resist these framings due to the specificity of intersex health and healthcare needs [13,14].

### 1.1. Intersex Variance

People with intersex variations are estimated to form between 1.7% to 4% of the general population [15,16,17]. People are born with or develop intersex variance as physical, hormonal or genetic features. These variations relate to a range of physical traits that lie outside the binary, medical or social norms of male and female [18]. Male or female representation of sex traits therefore remains problematic for people with intersex variance as they are too often confined, made invisible or problematised in this continuing use of dominant binary categories. Conceptualisations of intersex variation benefits from a non-binary understanding of sex that moves beyond male or female ideals [8,14]. Furthermore, naïve or simplistic readings or understandings of intersex variance are to be avoided. The complex politics of claiming an intersex identity is revealed in theoretical debates as this may include people who identify as intersex (as an identity category), without having the variation in biological physical, hormonal or genetic sex characteristics [4].

Thus intersex embodiment frames sex as a spectrum, rather than a binary, which then makes visible the many forms of intersex variance that exist. Even within biomedicine, genetics and human biology, scientists have argued for a more varied representation of sex and contemporary representations support these notions of human sex spanning beyond male or female binary sex, and basing representation on a more nuanced sex spectrum [19]. Given that a range of intersex variations exist, framing sex as a spectrum serves to further trouble the notion of apriori biologically determined sex and the stability of an individuated, liberal healthcare subject. However social pressure to conform to binary sex norms is immense [19].

Further work is needed to respect and promote the fundamental rights of people with intersex variance globally such as the right to bodily integrity and self-determination combined with the right to decline biomedical interventions where preferred [18]. People with intersex variance are more likely to report incidents of lifetime and everyday discrimination compared to the general population and may experience higher levels of unpredictable, episodic and day to day ‘minority’ stress because of discrimination and stigmatisation leading to social isolation and limited understanding of their lives by others. These factors can act as significant barriers where people with intersex variance access health and social care services [4,14,20]. Moreover, the need for research including people with these variations is widely reported within contemporary health research [4,20,21].

### 1.2. Intersex Healthcare

As the concept of intersex relates to variations that lie outside binary ideals of male and female, people with intersex variance may be exposed to biomedical interventions, including hormonal treatment or surgery on minors to align their bodies with typical male/female sex characteristics, with complex challenges around provision of informed consent. For some people with intersex variance these interventions may have lifelong consequences due to surgical scarring, the continuation of surgery into adulthood or the effects of trauma linked to surgery that combines with the emotional impact of discrimination and stigma. The following citation poignantly states: *I was sixteen when I first received an intersex diagnosis, though that wasn’t the terminology used. I had gone to a general practitioner—I hesitate to say “my” general practitioner, as regular physician visits were an irregularity in the hovering-around-the-poverty-line world of my adolescence—to find out why I hadn’t begun to menstruate. After multiple visits, bloodwork, and trips to specialists, I was told that I wouldn’t be able to have children, that my body needed a bit of a push if it were going to more adequately “feminize”* [22]. (Malatino 2019: p. 15).

Critical Intersex Studies and Critical LGBTI Liberationist lenses, for example, question the need for these processes of alignment and argue for non-normative formulations of human sex [4,20]. Furhtermore, ethical concerns persist in healthcare practice, particularly around lack of adequate information and lack of informed consent for practices ranging from anatomical photography through to application of so-called ‘normalising’ genital surgery on minors with intersex variance. Misconceptions of health professionals are often based on intersex embodiment being poorly understood as the assumption of binary male/female sex persists.

Evidence suggests that people with intersex variance are more likely than the general population to report unfavourable experiences of accessing healthcare including poor communication from health professionals and dissatisfaction with the treatment and care received [4,20]. Health professionals commonly accept male/female binary sex as a norm and express surprise when they learn of intersex variance [14], but feel unsure of how to accommodate the specific health needs of these individuals. Their health needs are often poorly understood by health professionals due to limited knowledge and understanding [2]. Opportunities for learning are inadequate as people may avoid disclosing their intersex variance mostly out of fear of curiosity or comments of health professionals [4,20].

Associated with healthcare, the lives of some people with intersex variance are unnecessarily medicalised via biomedical terminology describing intersex variations as ‘disorders of sex development’ (DSM), as differentiated in Figure 1.

An Australian study asked participants (*n* = 272) to self-identify and select from intersex variations that they were born with including more than 30 options ranging from: complete or partial androgen insensitivity syndrome (AIS); gonadal dysgenesis; congenital adrenal hyperplasia (CAH); 5α-reductase deficiency; 17β-hydroxysteriod dehydrogenase deficiency; aphallia; cryptorchidism; epispadias; Leydig cell hypoplasia; de la Chapelle syndrome; Denys-Drash syndrome with Fraser syndrome; hyperandrogenism; Jacobs syndrome; Kallmann syndrome; Klinefelter syndrome; Micropenis; Mosaicism involving ‘sex chromosomes; MRKH; Mullerian aplasia; Mayer-Rokitansky-Küster-Hauser (MRKH) syndrome; ovotesticular disorder; progestin-induced virilisation; Swyer syndrome; Turner’s syndrome etc.

However, more importantly, these diagnoses can be incongruous with how some people self-identify [8]. Although the benefit of biomedical assessment and related intervention for conditions such as Ullrich-Turner syndrome or Klinefelter syndrome may be acknowledged, activists representing this broader population group, drawing on a critical sociology health frame, present intersex variation as a biological difference instead of a medical pathology [4,24,25]. However intersex variations may be unnecessarily medicalised in healthcare settings where categorisations such as ‘disorders of sex development’ (DSD) persist [26]. Even though the World Health Organisation in 2018 replaced gender identity disorder with gender incongruence, understood as ‘a marked and persistent incongruence between a person’s experienced gender and their assigned sex’, these changes to the International Classification of Diseases (ICD-11) benefit trans and non-binary people only. Efforts continue to de-medicalise and de-pathologise ‘disorders of sex’ development for people with biological intersex variance, particularly where these terms are at odds with how people self-identify [27,28,29]. One helpful strategy exists in deconstructing these medical labels. The power deployed via the medicalisation of people with intersex variance can be reduced when diagnostic labels are interrogated, perceived as socially constructed terminology, utilised or withdrawn for strategic purposes either to access treatment or to avoid labelling and blend in [27].

## 2. Methods

The paper presents a review of the health and healthcare inequalities for people with intersex variation by addressing the following key question: What are the health inequalities and the experiences of accessing healthcare for people with intersex variance? The approach was selected to identify and consolidate research in the area by cutting across a developing field and representing multiple perspectives spanning biomedicine, health sciences and bioethics. Though the review may not deliver specific answers to given health concerns for people with intersex variance, the literature study may aid practitioners, policy makers and researchers to tackle problems bridging the field to inform education, policy and future research [3].

### 2.1. Search Strategy

Systematic searches were undertaken in five electronic databases [Medline (including PubMed), Web of Science, Cochrane Database of Systematic Reviews, PsycInfo, CINAHL]. Google Scholar was searched where further papers were identified in addition to reviewing the references for the selected papers. Searches undertaken in 2020 included results from prior searches conducted in 2016 for an associated project [8].

### 2.2. Inclusion Criteria

For the purposes of the review, people with intersex variance were studied as a whole group rather than referring to individual somatic intersex variations. Studies with participants who identified as intersex for political or gender identity purposes were excluded from the search. Where research included LGBTI (lesbian, gay, bisexual, trans and intersex) participants, only studies that made explicit reference to data collected with intersex participants were included. Although aggregating data for different groups can be useful for research impact purposes, it does blur important issues which may be particular to each group and merit specific attention [8]. As a result this review considered health and healthcare inequalities by actively prioritising people with intersex variance. Papers were reviewed for inclusion if they were: (i) primary qualitative, quantitative or mixed methods research; (ii) or systematic reviews or narrative synthesis. Papers were eligible if they were published in English with an accompanying full text for participants with intersex variance, without any age or geographical restriction. Grey literature, reports, unpublished research and theoretical papers were excluded.

Studies were included if published in peer reviewed journals within any setting and made available in the period between 2012 to 2019 to ensure currency of the included studies and following the UN affirming the legal obligations of ‘States to safeguard the human rights of LGBT and intersex people’ in 2012 [30]. This UN document ‘*Born free and equal: Sexual orientation and gender identity in international human rights law*’ made visible discrimination intersex people may experience, and affirmed that all people irrespective of sex, sexual orientation or gender identity are entitled to enjoy the protections provided for by international human rights law including in respect of rights to life, security of persons and privacy, the right to be free from torture, arbitrary arrest and detention, the right to be free from discrimination etc.

### 2.3. Key Search Terms

Search terms with alternative key terms in Medical Subject Heading (MESH) were utilised for the purposes of database search engines and included (1) ‘intersex *’ or ‘disorders of sex development’ or ‘hermaphrodit*’ and (2) ‘health *’ and (3) ‘healthcare’ or ‘health care’ and (4) ‘inequalit *’ or ‘disparit*’ or ‘inequit *’ or ‘determinant’.

### 2.4. Data Extraction

The initial search delivered 9181 papers with Figure 2 indicating how papers were chosen. Full texts of 74 papers were screened for possible inclusion with primary research papers (*n* = 14) and reviews (*n* = 2) included in the synthesis that follows. EndNote x9 software was utilised to extract papers from electronic databases by assessing the titles and abstracts for suitability. The identified papers were retrieved as full texts and screened where they met the inclusion criteria. These papers were independently extracted with references of included papers scanned by one reviewer, with the synthesis checked by a second reviewer.

Summaries of the features for the included papers combined with their findings were logged in Microsoft Excel (Table 1). Studies were appraised via the Critical Appraisal Skills Programme (CASP) framework for qualitative (Table 2), quantitative or mixed methods (Table 3), and review papers (Table 4) with the aim to follow a consistent approach across all studies. Where questions within the appraisal framework did not have a clear answer, for example assessing the value of the quantitative research (no 10), these questions were excluded.

### 2.5. Synthesis

The results for quantitative or mixed methods studies (*n* = 9), qualitative studies (*n* = 5) and reviews (*n* = 2) were presented as a narrative synthesis. Meta-analysis of the quantitative research could not be undertaken as studies included a range of different designs and outcomes measures. One researcher identified themes that were later verified by a second researcher. Quotes extracted from the research were available to reflect specific notions that emerged across data. Qualitative, quantitative and review results were combined to lend value to the body of evidence in its entirety.

## 3. Results

The results will present the review findings to address the health inequalities and the experiences of accessing healthcare for people with intersex variance. Research reporting on intersex health and healthcare is limited and often small-scale, ranging from one participant in Hughes [38], to 123 in Berglund et al. [36], and 329 in Bennecke et al. [35]. A synthesised summary of what is known follows.

### 3.1. Mental Health

An Australian survey of people with intersex variance (*n* = 272) aged 16 to 87 found being diagnosed with a ‘DSD’ or ‘disorders of sex development’ and the related medical intervention, had a range of both physical and psychological effects with young people experiencing isolation due to stigma, bullying, discrimination or rejection from family or peers [32]. Consequently 26% of people with intersex variations in the sample have engaged in self-harm with the incidence of suicide attempts were 19%. As many as 60% had considered suicide, compared to under 3% of the general Australian population [32]. An Italian cross-sectional study of individuals with 46,XY ‘disorders of sex development’ (*n* = 43) including Androgen insensitivity syndrome (AIS), complete gonadal dysgenesis (GD), 5α-reductase deficiency and Leydig cell hypoplasia, found higher rates of psychological distress including depression and anxiety in women with intersex variance amongst study participants compared to the general population [37].

However research indicated that support to address these problems were available in some settings. A study undertaken in Germany, Austria and Switzerland for people with intersex variance (*n* = 110), found in 11% of participants (*n* = 12) access to psychological support services in the form of counselling or talking therapies to help maintain mental health, significantly increased patients’ satisfaction with healthcare. Research recommends that individuals with intersex variance should have access to long term follow-up and mental health services as part of interdisciplinary care, to maintain mental health from childhood to adulthood [20]. Within these services that were effective, long-term follow up included assessment of psychosexual, emotional and social wellbeing [18,20].

### 3.2. Assigned Sex

For some people with intersex variance, binary notions sex may be challenging where their bodies spanned binary sex characteristics, with related implications for their gender identity. In the Australian survey of people with intersex variations (*n* = 272), 52% of participants reported having been assigned female sex at birth. Of these participants, the same proportion (52%) continued to use the same assignation at the time of the survey. Amongst all participants, 41% were assigned male sex at birth, with 23% continuing to use the same male assignation. However some participants assigned male sex at birth, may over time have changed their gender identity when they were older. In this same study, 8% of participants identified as being trans due to disagreeing with medical practitioners about their assigned sex [32].

### 3.3. Gender Identity

As seen above, in the Australian study where 41% of male participants (*n* = 272), assigned male sex at birth changed their gender identity over time, with 8% of the total participants later identifying as trans [32]. This study reflects the complexity of sex and gender, where individuals disagreed with medical practitioners’ assessment of their physical sex characteristics and changed their assigned sex as they became older, with related implications for gender identity. Where gender identities and gender roles of people with intersex variance fell beyond the gender binary, they represented neither male nor female gender roles, but instead wanting to ‘live as that which I am’. A mixed-methods study with people with intersex variance (*n* = 69) undertaken in Germany, found gender expression was more flexible and associated with the right to an ‘authentic and intersexual self’ falling beyond binary identity [41]. Study findings identified for the total population that gender allocation at birth was female in 83% and male in 17%, with 75% of people satisfied with their gender allocation. Of these people as adults, 81% lived in the female gender role and 12% living in the male role, with as many as 7% opting for other roles. Over time 9% reported a change in their gender. More significantly 24% reported an inclusive or mixed gender that combined male and female components, with 3% identifying as neither male nor female. Of the total participants (*n* = 69), 26% expressed significant uncertainty regarding belonging to a specific gender. The study highlights how binary categories of gender do not represent the gender for a substantial proportion of people with intersex variance who participated in the study that identified as mixed or both (24%), and neither or non-binary (3%). Findings support the need to reconsider sex [32], and gender [41], to reflect greater diversity by preventing people with intersex variance from being ‘trapped’ in, or reduced by these limiting categories. Further research recommends that the gender identity of young people should be respected including when they identify as non-binary, whilst approaching puberty and beyond [18].

### 3.4. Healthcare

Research found a range of healthcare associated concerns for people with intersex variance ranging from problematic sexual experiences after medical intervention [40], dissatisfaction with treatment and surgery [20], combined with sexual desire problems [18], and an absence of patient or service user consultation regarding their health needs [40]. Furthermore the views and choices regarding their own treatment preferences were not being heard by health professionals or taken into account during clinical decision-making [27]. More detailed accounts of these studies follow.

### 3.5. Surgery

Research findings challenged assumptions that early surgery to feminise or masculinise the bodies of people with intersex variance where there is difference, is in the best interest of these individuals [18,20]. In a European study with (46, XY) people with intersex variance (*n* = 57) who had undergone genital surgery, 47% were unhappy with the outcome of surgery, 70% had problems with sexual desire and 56% described symptoms of dyspareunia whilst 44% XY males feared sexual contact compared to 81% XY females. Overall dissatisfaction with sex life of XY females were 42% compared to 11% for the general female population [18]. As the negative impact of surgery has been highlighted for health and wellbeing, researchers recommend that early feminising surgery should be avoided at birth, and gonadectomy should only take place where there is a risk of gonadal malignancies and in consultation with parents along with the child or young person. In addition, non-emergency surgical intervention should be reduced to a minimum [41], and should only take place with full informed consent in accordance to patient’ needs preferably in puberty and adulthood [18,31,42]. More recent studies report a departure from performing unessential surgery on children and young people in Australia [32], and Germany [31,42]. Parents should be informed of the diagnosis of their child at the earliest opportunity and should be consulted with their child during decision-making about treatment options and care pathways. Where possible the child should be assigned to the most likely gender, and where this is not possible a decision to rear a child in a specific gender should be delayed and left open to ensure the family consult the child whilst their gender evolves [18].

### 3.6. Health Professionals

Health professionals held a range of views regarding intersex variation that are informed by biomedical knowledge and sociocultural norms. People with intersex variance are assessed in clinical healthcare settings where, what constitutes male or female bodies is subjected to biomedical scrutiny. Along these lines a qualitative study conducted in-depth interviews with medical professionals, parents and people with intersex variance (*n* = 62) in ‘DSD’ centres based in Israel and Germany respectively to understand how medical guidelines for treatment of ‘disorders of sex development’ shaped intervention. Where the bodies of young children were assessed based on sex binaries, the related treatment reflected the views of health professionals and whether intervention was required or not. Israeli practice included ‘normalising’ surgical intervention to align the bodies of young children to male/female sex characteristics, whereas German practice avoided unnecessary surgery, and instead offered parents psychological support to understand somatic variance [42]. The following quotations reflect a range of views expressed by health professionals in the two settings with significant implications for intersex children:


*… nature has decided (tapping the table) that there should be a female and a male and when they are together then you can have a baby, yes? There is a God and he decided that it should be this way. With human beings and with all animals it’s the same: there are female and male animals.*
[42] (Prof. K., 29 October 2015 in Meoded Danon 2019 p. 150).

Based on social and cultural norms, this quote reflects the engrained binary notion of biological sex that fails to take account of variation in sex characteristics that moves beyond or exceeds male and female sex.


*…First, when these children are born, they have genitalia that are neither male nor female, and society cannot accept this. This family seems bizarre to society. How will they put him in a kindergarten? Second, from a technical point of view, healing is faster, and there is a great advantage in doing these operations at a young age. Third, psychologically, a girl [in this case] also has to grow up knowing that she is more or less normal. She will grow up with female genitalia and not some intermediate thing that appears to be something more masculine. So, the goal is to fix these children between the age of six months and a year. This is the optimum age.*
(Prof B., 25 Oct 2015 in Meoded Danon [42] p. 156).

The above quote reflects the normative ideal based on male and female sex with the bodies of those who do not fit the binary needing to be ‘fixed’. We could ask, are these bodies ‘broken’? If they are fixed what are the consequences for those who did not consent to invasive medical intervention? Where intervention is not essential, if the binary notion of sex was disrupted, could sex be reframed where more diverse sex characteristics are accounted for?


*I say what I always say, which is ‘we are happy that your child is healthy. We don’t need to do anything. Everything is fine. Look at your beautiful child’. I will say this several times, so it is clear that there is no deal [no rush to do anything] now. So they will not do anything, I tell them what I think is most important so they can connect to their child.*
(Dr. E., 28 November 2015 in Meoded Danon [42], p. 155).

In this quote the health professional puts parents at ease by laying the emphasis on the ‘health’ and ‘beauty’ of the child instead of medicalising what is in essence a biological variation in sex characteristics.


*We really want to avoid early operations. We seriously spend a lot of time talking about the need to open our minds to the fact that we have more than two “drawers”, boys or girls, that it is a task for society to see the whole spectrum, that it is a challenge for the family to understand that this child is special and that we should wait until the child is able to express an opinion with his own voice. So regarding everything that has to do with correcting the genitalia we are very, very, very careful.*
(Dr. M., 8 April, 2016 in Meoded Danon [42], p. 155).

The above quote reflects the notion of sex as a spectrum that incorporates a range of sex characteristics beyond male and female. When this view is held by health professionals, the need for early unnecessary intervention is reduced. These quotations show how sociocultural norms of sex and gender inform biomedical practice around the absence or presence of surgical intervention for children with intersex variance in different international healthcare settings. With adequate specialist treatment that includes psychological support for parents, early unnecessary surgery can be avoided or delayed until young people are able to decide for themselves and provide informed consent for invasive treatment options. Where health professionals and parents understood sex as a spectrum, children were approached with appreciation for their variance. These views afforded children greater flexibility as their gender identity evolved over time.

### 3.7. Accessing Healthcare

Communication between people with intersex variance and those involved in their care and socialisation had a significant impact on psychological wellbeing. Poor communication between health professionals, the family and the patient combined with secrecy or stigma related to people with intersex variation, adds to the psychological burden of these conditions [20,31]. UK based research with young women based on their intersex variance (*n* = 13) aimed to understand their experiences of sharing information with health professionals, friends and partners. The research found that where health professionals feared adverse responses from young women, they were less likely to disclose medical information. Improved communication and support may be needed following healthcare consultations where important information is shared by health professionals such as young women with intersex variance learning they may not be able to have children [38]. Improved communication with greater psychosocial support could help young people understand their intersex variance, particularly where they may experience adverse reactions from others following disclosure of sensitive personal information [32].

In the Thyen et al., study with German, Austrian and Swiss participants (*n* = 110), 28% of those who participated experienced difficulties whilst accessing specialist care. Participants who did not understand the diagnosis at the time of disclosure reported significantly lower levels of patient satisfaction compared to the average satisfaction for all patients. Of the study sample, 11% of intersex participants reported being offered counselling or talking therapies to help them make sense of events in adulthood. 28% reported that they had been offered such services but that they had no need for them, however the majority of participants had never been offered access to psychological support services. Those who had never been offered psychological support reported the lowest satisfaction with care. Thus access to psychosocial support services appears to increase patient satisfaction with healthcare [20].

### 3.8. Parenting

In the German DSD Network study with children who have intersex traits aged 8–12 (*n* = 86), parents of these children expressed significant uncertainty of when and how to disclose or speak to their child about their intersex variance [39,43]. Related research later evaluated the clinical outcomes for individuals with ‘DSD’. The study included children and adolescents with their parents (*n* = 439) based in Austria, Switzerland and Germany. Research found a significant need for psychological support in parents of young people with intersex variance to help them consider the gender issues of their child where they may exist, and to reduce parents’ fears of stigmatisation [35]. Positive outcomes were noted for young people with intersex variance in settings where parental support groups were available. These groups aided parents to support their children and adolescents whilst they attempted to understand and make sense of their bodily variation [42].

### 3.9. Age

A number of studies included young people, for example the Italian study with adult females with 46, XY DSD (*n* = 43) who were accessing treatment with a comparison group of 46, XX healthy women (*n* = 43) found younger women with intersex variance showed better psychosocial adjustment and quality of life in the social arena comparand to older ones [37]. For this study psychological support was offered to 39% of the entire sample of 43. Of these 46% considered the support useful. 55% of younger women were offered psychological support compared to 47% of the older subgroup (*p* = 0.041). Younger individuals required psychological care more often and earlier in the younger group than in the older group (70% vs. 35% respectively; *p* = 0.034). Younger women were informed of their condition significantly earlier that those who were older (mean age 16.5–4.2 vs. 21.2–8.6; *p* = 0.048). Furthermore younger women showed significantly better quality of life in the social arena (66.6–75 vs. 66.7–58.3; *p* = 0.003). However, less research is available for people with intersex variance who are older. An Australian survey with older LGBTI people (*n* = 312) including an intersex participant, highlighted a dearth in research with very few studies addressing the mental health, social support and loneliness of people with intersex variance as they age [38].

### 3.10. Social Factors

The European study that evaluated the surgical outcomes and sexual life of (XY, DSD) people with intersex variance (*n* = 57) found for younger people their peer, romantic or sexual relationships were challenging areas of life, leading to high levels of sexual anxiety [18]. Related findings of a German clinical evaluation that assess psychosexual development in adolescent (*n* = 66) and adult (*n* = 110) individuals with intersex variance, found that for adults, 25% never had a romantic or intimate relationship [39]. Guarding against disclosing personal information was seen as a form of self-preservation or protection against potential hostile reactions [34]. The following quote of a young woman with intersex variance reflects difficulties around intimate relationships.


*It’s hard for people like me to get a steady functional intimate relationship. Because if you think about it you’ve got your medical problems to explain for one and a lot of people these days are only out for one thing (sex) and you can’t just tell everyone that you can’t, things get round too fast. You have got to be careful. Everyone says you can have a decent life without a relationship, I don’t think that’s true. I’m sure there is, [someone like me] somewhere out there but I wouldn’t even speak to them about it all, because it’s just too awkward*
(a young woman in Sanders et al. [34], p. 1909).

Alongside challenges with social relationships, broader socioeconomic aspects may be influenced for people with intersex variance. A Danish nationwide registry study with 46, XY females (*n* = 123) including Androgen insensitivity (AIS), GD, 17α -OHD, 17β-HSD and Star mutation described the morbidity, mortality and socioeconomic status of these women in comparison to the general population during the period from 1960 to 2012. Participants were compared to a randomly selected aged-matched control cohort of 12,300 females and 12,300 males in the general population. Where the education, income, cohabitation, motherhood and retirement of people with intersex variance were compared to the general population, cohabitation and motherhood including the prospect of family life were reduced for females with intersex variation. However, education was similar to the general population, or slightly higher achievement was found for education and professional life for those with intersex variation over time [36].

Life story research based in Thailand with LGBTI people (*n* = 19) including a participant with intersex variation (*n* = 1), reported that experiences of discrimination may lead to unemployment and lower socio-economic status in a national setting where there was less or no legal protection in the workplace [33]. Conversely the Danish study reported that income for females with intersex variance was higher compared to the general population where they performed well in the labour market [32]. Thus socio-economic status for people with intersex variance varied across geographical areas depending on the level of income and whether legislation protected people against discrimination in the workplace [33,36].

## 4. Discussion

### 4.1. Summary of Findings

This review consolidates research undertaken in biomedicine, health sciences and bioethics and evidences a range of health inequalities for people with intersex variance. These inequalities can be seen in increased levels of psychological distress such as anxiety and depression compared to the general population or social isolation, stigma, discrimination and/or rejection from others. More positively, the availability of psychological support and counselling services increased patient satisfaction for both young people, adults and their parents, particularly where parents were uncertain of how to speak to their child about their intersex variance, or where they feared stigmatisation when the gender identity or assigned sex of the young person did not reflect binary male/female ideals. Improved communication with greater professional support can help young people to understand their own intersex variance in situations where they may experience adverse reactions from others following disclosure of sensitive personal information. Peer, romantic or sexual relationships were perceived as challenging, with many individuals with intersex variance avoiding romantic or intimate relationships as a form of self-preservation or as protection against potential hostile reactions from others. Cohabitation and motherhood, including the prospect of family life, were reduced for people with intersex variance compared to the general population. The socio-economic status for people with intersex variance varied across geographical settings depending on a range of factors including the level of legal protection against discrimination in the workplace.

Where sex norms persisted, these binary ideals were reflected in the views of health professionals and the treatment guidelines they followed. A range of intersex variations are diagnosed biomedically which continue to unnecessarily medicalise people with intersex variance based on their somatic variance. These terms are often incongruous with how people with intersex variance define themselves. This is reinforced in current framings of intersex variance as ‘disorders of sex development’ persist in systems of classification such as the WHO International Classification of Diseases (ICD-11) or the APA Diagnostic and Statistical Manual of Mental Disorders (DSM-V). The impact of this medicalisation of intersex variance is reduced when diagnostic labels are perceived as socially constructed terminology, utilised or withdrawn by people with intersex variance for strategic purposes either to gain access to biomedical treatment or to blend in. Where descriptions of bodily variation in pathological terms persist, surgical intervention informed by biomedical knowledge and sociocultural norms continue in international healthcare settings. Some research on the health of people with intersex variance relates to surgical intervention dedicated to assigning sex within the male/female binary, sometimes without gaining consent where surgery concerns minors with intersex traits. Promising practice is evident in specialist treatment centres that includes psychological support, where unessential surgery is delayed or avoided until people with intersex variance can self-identify or provide fully informed consent to undergo invasive treatment options. In settings where health professionals resisted binary notions of sex, people were approached with appreciation for their somatic variance, whilst their gender evolved over time. Review findings support the need to rethink sex and gender to reflect greater diversity, particularly where people identify as non-binary within the gender-sex spectrum.

### 4.2. Strengths and Limitations of the Review

With regards to the methodological approach followed for the review, the lack of independent screening and extraction of papers in duplicate would have increased rigour and is therefore regarded as a limitation. A further limitation is not undertaking the study with a prospective protocol. Prior registration of a prospective protocol with Prospero would have aided the review. A number of studies included people with intersex variance in research with lesbian, gay, bisexual, trans and intersex groups [1,33,38]. Studies combined groups often within rights, legal or policy considerations to maximise visibility and collective impact [8]. Although aggregating data for diverse groups can be useful for analytical purposes and research impact, it does blur important issues which may be specific to each group and their related health or healthcare needs [4]. Therefore by recognising the merits of research separating groups into more nuanced component parts and disaggregating data for people with intersex variance in particular, this review provides a synthesis of health and healthcare research for people with intersex variance as a distinct group. Furthermore this review differs from prior studies that either focused on health inequalities of LGBTI people [1,8], or explored conceptual or theoretical lenses for representation of intersex health more broadly in a review of research including grey literature from 2015–2016 [4]. This is a strength of this study insofar as expanding knowledge by drawing on existing studies in the field, and adding to content with a more defined critique of research and an accompanying focus on the healthcare inequalities of people with intersex variance.

With the exception of Chinese, Israeli and Thai studies included in the review, research represented was mainly undertaken in Australia, Europe, USA, Switzerland and the UK. As a result the literature can be regarded as biased insofar as representing largely Western views. Future reviews should include more nuanced representation for a range of geographical settings to reflect varying sociocultural and legal factors globally that will shape the experiences of people with intersex variance where they access healthcare.

### 4.3. Implications for Research

The review found a lack of substantive research on the general health and cancer burden of people with intersex variance. Further large-scale research is needed to understand the general health profile for those with intersex variance and their experiences of accessing healthcare. Very few studies address mental health [44], or the psychological wellbeing and social support of people with intersex variance as they age [4,45]. Thus limited information is currently available for this group where gender, sex and age intersect.

Research will benefit from theoretical underpinning by emergent theories with a specific focus on the material and embodied realities of people with intersex variance. Theory informed research will raise crucial questions around how to represent and empirically investigate identities and subjectivities that are in a state of flux and constantly evolving with greater ethical accountability [46]. In health research, how do we meaningfully investigate the embodied processes for people with intersex variance? Research is required to explore:How meanings or understandings of intersex subjectivity and embodiment materialise as multi-layered forms of interdependence via health assemblages.How to achieve dynamic systems change via disruption of biomedical linguistic essentialism (as seen in ‘DSD’ diagnoses), whilst questioning sex and gender normativity in healthcare.What these forms of resistance might look like.

Research informed by theory should be undertaken with a specific focus on intersex health in greater consultation and peer involvement of people with intersex variance, as they understand their own health needs and experiences of accessing services. Where this is unattainable and LGBTI research includes people with intersex variance, their goals should be actively prioritised in the research [4]. Co-producing research with peers who have experiential knowledge and lived experience should be a priority, by involving people with intersex variation at every level from conception, design to undertaking the research [2]. This kind of research should co-constitute materiality with more intersubjective methodologies and ethically accountable healthcare practice.

## 5. Conclusions

Given continuing biomedically informed understandings of sex variation, it is no surprise that restrictive clinical conceptualisations of the health or healthcare for people with intersex variance persists within health sciences. This medicalisation of the bodies and lives of people with intersex variance are, for some, accompanied by damaging consequences and disparities with lived subjective experience. For others, this provides much needed access to treatment, hormonal and surgical intervention. Although substantial contemporary research is concentrated within biomedicine or health sciences and bioethics, there is some critical, patient-centred, community driven and theory informed work becoming visible. Emergent thinking and activism reflected in health research is marking a shift in the way intersex variance is framed by drawing attention to the ethical and political implications associated with such healthcare practices. Where practice is based on the continuation of normative gender and sex binaries, combined with the pathologising language of biomedicine, discursive, embodied and material tensions become more apparent. Both politically and ethically, these familiar normative binary sex and gender framings remain in tension with the equal rights afforded through equalities legislation and in settings where gender and sex are protected characteristics. However further work is needed to ensure equalities legislation takes account of more varied sex characteristics that span the binary male or female spectrum. Political questions and challenges to unwanted medical and healthcare interventions, and as importantly, demands for greater ethical accountability from healthcare professionals, are being voiced and heard. Much scope remains to address the general health profile and the subjective embodied experiences of people with intersex variance when they access healthcare. Research should be coproduced alongside peers with intersex variance, including those who are older, to inform practice, research and policy. Co-production or coproduced research in healthcare is an approach that brings together the public, practitioners and researchers to generate more meaningful knowledge. This is research that starts from the premise of inclusivity of all knowledges, skills and perspectives, is based upon values of respect and reciprocity, and most importantly, fully considers question of power, with the sharing of and joint ownership of research [47,48]. These levels of recognition throughout the research process, and especially in healthcare research, are vitally important for people with intersex variance, given the evidence presented here of ongoing objectification, misrecognition, marginalisation or exclusion. Coproduced research generating awareness of subjective lived experiences and learning from intersex activism will enable more acceptable, responsive and appropriate research which can lead to more inclusive and affirming encounters with healthcare. Central to this, and any healthcare practice, will be the need to interrogate, undo and rethink sex and gender to make visible and fully recognise greater diversity. Healthcare may then become more authentically appropriate and accessible to all within the gender and sex spectrum.

## Figures and Tables

**Figure 1 ijerph-17-06533-f001:**
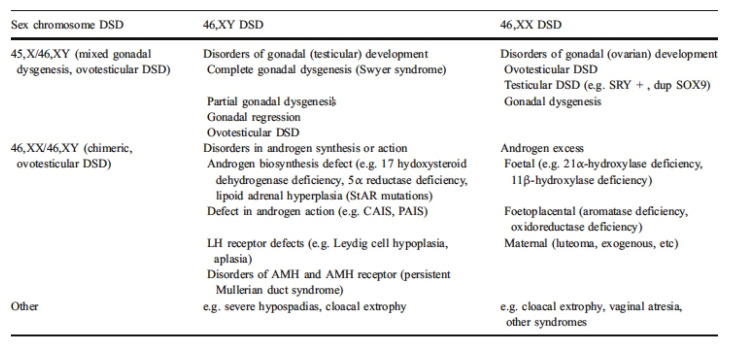
Example classification for ‘disorders of sex development’ [23].

**Figure 2 ijerph-17-06533-f002:**
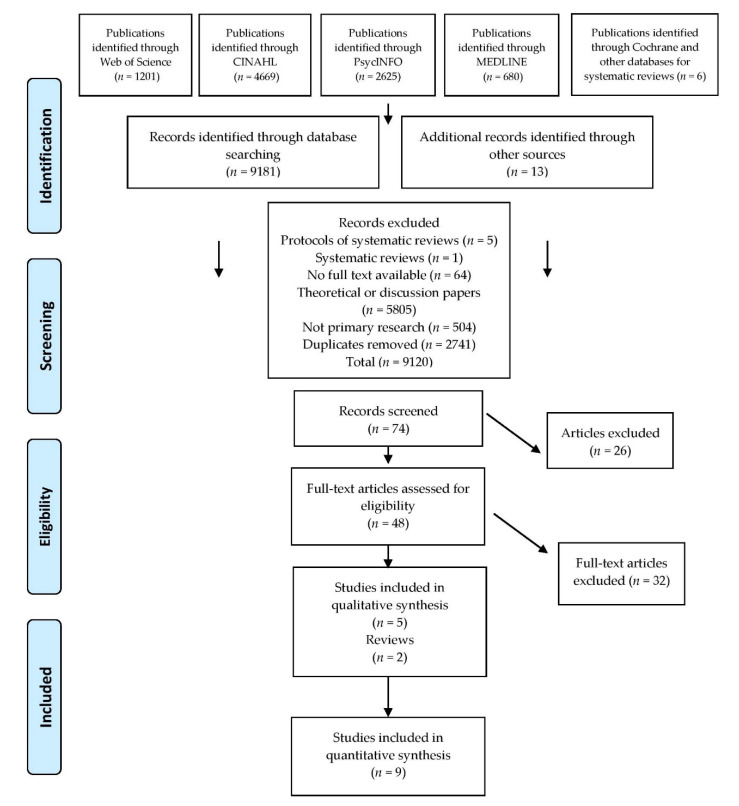
Preferred Reporting Items for Systematic Reviews and Meta-Analyses (PRISMA) flow diagram.

**Table 1 ijerph-17-06533-t001:** Characteristics of included studies.

Authors, Year	Country	Design	Population	Participant Numbers	Outcome of Interest
Bennecke et al., 2015	Germany, Austria and Switzerland	Multicentre clinical evaluation with ‘dsd’ questionnaire and SPSS statistical analysis	Young people diagnosed with ‘dsd-XX or XY without (c) or with partial (p) androgen effects, and female (f) or male (m) sex of rearing: dsd-XX-p-f, dsd- XY-p-f, dsd-XY-p-m and dsd-XY-c-f.	children and adolescents (*n* = 329) and parents (*n* = 110)	(1) Parents of children with ‘dsd’ have a significant need for psychological support. (2) Half of parents did not receive support. (3) Support should be part of multidisciplinary care of parents to reduce their fears of stigmatisation and to discuss gender issues, hormonal treatment and surgery of their child.
Berglund et al., 2018	Denmark	Nationwide registry study with random selection of participants and a control cohort with statistical analysis	46,XY females with Androgen insensitivity (AIS), GD, 17α -OHD, 17beta-HSD, WT-1 and Star mutation	XY females (*n* = 123), with a control cohort of females (*n* = 12,300) and males (*n* = 12,300) from the general population	(1) Mortality and education were similar to controls. (2) Cohabitation and motherhood were reduced for XY females compared to the controls. (3) Income and performance in the labour market were higher amongst in XY females later in life compared to the general population.
D’Alberton et al., 2015	Italian	Cross-sectional study with standardised (ABCL, WHOQOL) questionnaires	46,XY females with ‘dsd’ with AIS, Gonadal dysgenesis, 5α -reductase deficiency, Leydig cell hypoplasia	(*n* = 43) aged 18–57 years	(1) Statistically higher scores than the comparison group for depression, anxiety, internalising and externalising problems. (2) Younger people were more likely to access psychological support. (3) Lower psychological distress in younger women could indicate positive outcomes of changes in management.
Davis 2014	USA	In-depth individual interviews standpoint feminist analysis	Individuals with intersex traits	(*n* = 37)	(1) ‘dsd’ terminology is received and utilised in different ways, embraced by some and refuted by other participants. (2) Self-understanding might conflict with the ‘dsd’ terminology ascribed in the 1990s.
Meoded Danon 2018	Israel and Germany	Narrative interviews	People with intersex variations, medical professionals, parents	Total (*n* = 62) German adults with intersex variations (*n* = 4), parents (*n* = 4), professionals (*n* = 18); and Israeli adults with intersex variations (*n* = 15), professionals (*n* = 34), parents (*n* = 11)	(1) Health professional expressed range of views regarding people with intersex variance, sex as a spectrum, treatment options and whether intervention is needed or not. (2) The importance of intersex children meeting each other for support. (3) Legal reform can help prevent unnecessary surgery on minors with intersex variance.
Hughes 2018	Australia	Survey (SF12 and with a Likert scale and Kessler 10) with statistical and bivariate analysis	LGBTI older people	LGBT people (*n*-312), older adult with intersex variance (*n* = 1)	(1) Though many older LGBTI people are well, both physically and mentally, they do appear to face increased risk of certain health issues compared with the general population such as loneliness and psychological distress. (2) Participants with intersex variation were underrepresented in the sample.
Jones 2016	Australian	Survey with a descriptive comparative statistical analysis	People with intersex variations	People with intersex variance (*n* = 272) aged 16 to 87 years	(1) 42% of participants thought about self-harm on the basis of issues related to having an intersex variation (2) 26% had engaged in self-harm on the basis of having an intersex variation. (3) 60% had thought about suicide, and 19% had attempted suicide compared to under 3% for the general Australian population.
Jones 2018	International	Systematic review of health literature via theoretical lenses	People with intersex variations	Studies published in 2015–2016 (*n* = 61)	(1) studies framing medical interventions as problematic (*n* = 27). (2) studies using a bioethical/narrative inquiry frame (*n* = 26). (3) Studies with clinical medical theoretical lens (*n* = 24). (4) Studies with a critical LGBTI liberationist theoretical lens (*n* = 4).
Jurgensen et al., 2013	Germany, Austria, Switzerland	Questionnaire via interviews with statistical analysis	People with ‘dsd’	Adolescents (*n* = 66) and adults (*n* = 110)	(1) Partnership and sexuality were identified as challenging areas of life. (2) Fewer experiences of peer, romantic or sexual relationships were reported compared to the general population. (3) 25% of adults with ‘dsd’ never had a love relationship and will benefit from support and counselling.
Köhler et al., 2012	Germany, Austria, Switzerland	Evaluation via questionnaire with statistical analysis	Individuals with ‘46,XY,dsd’	People with intersex variance (*n* = 57) aged 18–62	(1) Constructive genital surgery should be minimised and only undertaken with informed consent mainly in adolescence or adulthood. (2) Multidisciplinary care and psychological support should include parents, peers and patient groups.
Ojanen et al., 2018	Thailand	Life story interviews with thematic analysis	LGBTI people	A person with intersex variance (*n* = 1)	(1) Transgender and intersex participants reported more discrimination and exclusion compared to LGB people. (2) Lower socio-economic status of LGBTI people results in vulnerability.
Thyen et al., 2014	Germany, Austria, Switzerland	Clinical evaluation via cross-sectional assessment with statistical analysis in SPSS	People with intersex variations	Adults with ‘dsd’ (*n* = 110)	(1) People with intersex variance should have access to mental health services as part of interdisciplinary care. (2) long-term follow-up should include measures of satisfaction with care and subjective psychosexual, emotional, and social well-being.
Sanders et al., 2015	UK	Interpretive phenomenological analysis	Young people with intersex variations	Young women with ‘dsd’ aged 14–19 (*n* = 13)	(1) Young women may fear sharing personal information (2) Physical intimacy may require planning which has an impact on their perceived expectation of sexual spontaneity in a relationship. (3) For those who can’t have children meaning given to fertility change over time.
Schweizer et al., 2014	Germany	Questionnaire, standardised scales, qualitative content and statistical analysis	People with intersex variations	Young people and adults with ‘dsd’ (*n* = 78)	(1) 24% of participants reported an inclusive two-gender/mixed identity and 3% neither male nor female gender identity. (2) Uncertainty of belonging to the female or male gender category as well as non-binary identifications highlight the need for alternative gender categories.
Wang and Tian 2015	China	Case-control with SPSS statistical analysis	Patients with ‘dsd’	People with ‘dsd’ (*n* = 87) aged 13–38	(1) Only 13.7% of participants partook in sexual activity. (2) Quality of life of ‘dsd’ patients are not significantly lower compared to the urban Chinese population.
Zeeman et al., 2018	International	Narrative synthesis of systematic reviews, meta-synthesis and primary research	LGBTI people	Studies of people with intersex variations (*n* = 8) published 2010–2016	(1) A significant lack of research exists on the general health profile and healthcare experiences of intersex people. (2) Unessential corrective surgery on intersex minors to align their bodies to the male/female binary should only occur when the young person can provide informed consent. (3) Intersex variations are diagnosed biomedically which unnecessarily medicalises intersex people. (4) Male/female binary categories for sex markers and gender identify are not helpful as intersex bodies can be ‘trapped’ in these limiting categories.

**Table 2 ijerph-17-06533-t002:** Critical Appraisal Skills Programme (CASP) quality assessment of qualitative studies.

No	Study	1	2	3	4	5	6	7	8	9	10
1	Davis 2014 [27]	y	y	y	y	y	y	y	y	ct	y
2	Meoded Danon 2018 [31]	y	y	y	y	y	y	y	y	ct	y
3	Jones 2016 [32]	y	y	y	y	y	ct	y	y	y	y
4	Ojanen et al., 2019 [33]	y	y	y	y	y	y	y	y	y	y
5	Sanders et al., 2015 [34]	y	y	y	y	y	ct	y	y	y	y

Checklist questions were: 1. Was there a clear statement of the aims of the research? 2. Is a qualitative methodology appropriate? 3. Was the research design appropriate to address the aims of the research? 4. Was the recruitment strategy appropriate to the aims of the research? 5. Was the data collected in a way that addressed the research issue? 6. Has the relationship between researcher and participants been adequately considered? 7. Have ethical issues been taken into consideration? 8. Was the data analysis sufficiently rigorous? 9. Is there a clear statement of findings? 10. How valuable is the research? Abbreviations: Y—yes; CT—cannot tell; N—no; N/A—not applicable.

**Table 3 ijerph-17-06533-t003:** Critical Appraisal Skills Programme (CASP) quality assessment of quantitative studies.

No	Study	1	2	3	4	5a	5b	6a	6b	9	10	11
1	Bennecke et al., 2015 [35]	y	y	y	y	ct	ct	y	ct	y	y	y
2	Berglund et al., 2018 [36]	y	y	y	y	ct	ct	y	y	y	y	y
3	D’Alberton [37]	y	y	y	y	ct	ct	y	y	y	y	y
4	Hughes 2018 [38]	y	ct	ct	ct	y	y	ct	ct	y	ct	y
5	Jürgensen et al., 2013 [39]	y	y	y	y	y	y	ct	n/a	y	y	y
6	Köhler et al., 2012 [18]	y	y	y	y	y	y	ct	ct	y	y	y
7	Thyen et al., 2014 [20]	y	y	y	y	y	y	y	y	y	y	y
8	Wang and Tian 2015 [40]	y	y	y	y	y	y	ct	ct	ct	ct	ct
9	Schweizer 2014 [41]	y	y	y	y	y	y	ct	y	y	y	y

Checklist questions were: 1. Did the study address a clearly focused issue? 2. Was the cohort recruited in an acceptable way? 3. Was the exposure accurately measured to minimise bias? 4. Was the outcome accurately measured to minimise bias? 5a. Have the authors identified all important confounding factors? 5b. Have they taken account of the confounding factors in the design and/or analysis? 6a. Was the follow up of subjects complete enough? 6b. Was the follow up of subjects long enough? 9. Do you believe the results? 10. Can the results be applied to the local population? 11. Do the results of this study fit with other available evidence? Abbreviations: y—yes; ct—cannot tell; n—no; n/a—not applicable.

**Table 4 ijerph-17-06533-t004:** Critical Appraisal Skills Programme (CASP) quality assessment of reviews.

No	Study	1	2	3	4	5	6	7	8	9	10
1	Jones 2018 [4]	y	y	ct	n/a	n/a	n/a	ct	y	y	n/a
2	Zeeman et al., 2018 [1]	y	y	y	n/a	y	y	y	y	y	n/a

Checklist questions were: 1. Did the review address a clearly focused question? 2. Did the authors look for the right type of papers? 3. Were all the important, relevant studies included? 4. Did the authors assess the quality of the included studies? 5. If the results of the review have been combined, was it reasonable to do so? 6. What are the overall results of the review? 7. How precise are the results? 8. Can the results be applied to the local population? 9. Were all important outcomes considered? 10. Are the benefits worth the harms and costs? Abbreviations: Y—yes; CT—cannot tell; N—no; N/A—not applicable.

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
