# Peer review of "A Systematic Review of the Health and Healthcare Inequalities for People with Intersex Variance"

_ijerph, 2020, doi:10.3390/ijerph17186533_

Round 1

Reviewer 1 Report

Summary

This systematic review by Zeeman and Aranda explores healthcare inequalities in individuals with intersex variance. The study protocol sought to identity all peer-reviewed articles that included assessment of individuals identified as having intersex variance, and compare a variety of health and healthcare factors. Multiple databases were included in the search, including those typically used for medical, mental health, and ethical research. A total of sixteen papers were included in this review, including qualitative, quantitative, and narrative studies. The study authors determined that those with intersex variances experience differences in gender identity, overall health, mental health, socioeconomic health, access to healthcare, and both personal and professional relationships compared to their peers without an intersex diagnosis. They furthermore identified several factors that were associate with both positive and negative outcomes in these areas. They recommend that additional research be performed to further explore these issues, and identify areas where identified barriers can be overcome to improve the overall health of this population.

Major Comments

  1. The terms “intersex people,” “intersex minor,” “intersex women,” “intersex participants,” “intersex bodies”, “intersex peers,” etc are used frequently throughout the manuscript. It would be more inclusive and less jarring to refer to these individuals as “people with intersex variance,” and “individuals with intersex variance,” as is also done in the paper. Avoiding defining people by their diagnosis is good practice.
  2. I would caution the authors to avoid blanket statements such as that in lines 116 and 121, stating that those with intersex variation are unnecessarily medicalized, and that this diagnosis should be separated from an ICD-11 or similar diagnosis. There are several intersex conditions, such as Turner syndrome and congenital adrenal hyperplasia, that do require medical assessment, sometime urgently and/or emergently, and identifying those who have these conditions could be life-saving.

Minor Comments

  1. In line 32 the authors refer to disease burden, but do not define to specific disease or health care concern they are referring.
  2. Rather than using the terms “non-normative” and “whole male/female” in referring to the bodies of those with intersex variance, consider terms such as atypical (lines 51 and 52).
  3. Lines 122-126. Although it is true the ICD codes changed for those who identify as transgender, again, we must be careful to ensure that the distinction between gender identity and physical sex is made. Gender identity is not a medical condition that could result in harm if not recognized, while several intersex conditions can be.

Author Response

Thank you for the helpful comment and feedback from the three reviewers. We are pleased that all reviewers believe the paper should be published and we found their views useful and constructive. We have responded to their comments in the attached document, and have addressed their suggestions as far and as fully as possible in the revised version of this text. We hope this revised version is acceptable and eagerly look forward to hearing the outcome. 

Reviewer 2 Report

 I will arrange my review in two sections, the first are points that I really do think need addressing/changing. The second areas that I offer to you as things to consider but that you may or may not think need changing.

So, firstly that areas I think really do need addressing.

On page 2 I am very concerned that your opening arguments around intersex variance came across as taking a rather uncritical approach to the gender binary, in particular your phrase “neither wholly female nor wholly male” could be read as tacitly accepting that some people are wholly female or male. Later on in this paper its abundantly clear you are not trying to argue this, but I worry that many interested readers will be put off on page 2 if they feel an uncritical approach is being taken to these issues. I think you can likely make minor changes to the paragraph “Intersex variance” by using phrases such as “supposed” or “so called”, and generally making it clear from the outset that you are not accepting the gender binary as a normative framework against which intersex people are then measured as pathological. You may also like to make reference to gender theorists who suggest no-one is ever fully male of female. I think you cover a lot of this in the first paragraph of page 3 headed “Intersex Health”, I think a lot of this content is dealing with what is intersex, its only really the final paragraph after the quote that deals with health. I think this conceptual development and clear critical stance needs to come upfront in the paper. So maybe consider moving some content from page 3 to page 2. To be clear, I do not think there is anything conceptually wrong with your discussion of intersex as a category when the whole paper is taken into account, I just think the critical edge isn’t clearly expressed from the start. It needs to be to avoid people putting the paper down at page 2.

I also think this section is the logical place to address one of my other points, throughout the paper you make reference to several specific genetic/hormonal diagnoses, I think these need defining. I understand what most of them are, but on page 6 I’m afraid the first paper’s population description eludes me. It may work best to do this in a table, so all the possible definitions that you mention in the paper are collected in a clear manner. Again, this may help bolster the discussion around gender and sex being highly fluid, complex and contested.

Next I’m afraid I simply can’t work out how on page 5 (prisma diagram) you get from 2741 to 74 papers. All the other steps are clear, but this one isn’t. I think a sentence is needed to clarify.

Ok, onto the more minor suggestions that as I say, are offered as considerations rather than anything stronger.  

Firstly, I think including covid in the opening sentence doesn’t really work, there aren’t any clear conceptual links, so I would consider redrafting that to ‘grab’ the reader in a different way that may appeal in a more long-term manner. As a systematic review, I feel this is the sort of paper that will still be  read in 5-10 years time, and covid may not have the same “grab” by then.

Next, I think you need to state your review question somewhere on page 4 rather than on page 11.

Regarding limitations, it may be worth reflecting on the (largely) Western nature of the identified literature. Why may this be? Do other cultures integrate/oppress intersex people in ways that mean it can’t be or doesn’t need to be researched in the same way? Even just noting the racial bias to the literature could be enough.  

Page 13, you have some really interesting quotes from the literature that suppose and oppose surgery for children. I think there could be a clearer critique of the supporting, and then opposing quotes, followed by a concluding paragraph. There’s some fascinating ideology being expressed here in terms of “having a baby” (is that all women are useful for?) and “fixing” (are they broken?) and equally discussion of “beauty” and “spectrums” I think could be picked out in more detail. You may of course wish to unpack different phrases and words to the ones I identify here, but is just seems like these were really rich quotes that weren’t made the most of.

All that being said, I do think this is an excellent example of a systematic review which with a few minor changes will really engage a wide audience in a constructive manner.

Author Response

(The authors gave the same response as above.)

Reviewer 3 Report

Thanks for the opportunity to review this evidence synthesis on a very important topic. The systematic review seems to be methodologically sound and the results and discussion are very interesting but there are some aspects of the study that could be reported with more transparency. I will list these below:

  • Why was research before 2012 excluded? It is difficult for readers to judge whether this was an appropriate decision when no justification is presented.
  • What distinguishes this systematic review from the one conducted by Jones (2018)? How might they complement each other? This would be useful information for readers.
  • An appendix with the full search strategy for at least one of the databases should be included and I think ay difference between the search strategy for this study and the related search in 2016 should be highlighted.
  • Why was grey literature excluded? So much research is now made available as a pre-print or, in this area of research in particular, as a report commissioned by a community organisation or public institution that it is vital to include these sources in systematic reviews.
  • If the full-text of an included paper was not available through your institution, what did you do?
  • There is mention of a protocol for this review but it is not cited. It should be.
  • The lack of independent screening and extraction in duplicate should be listed as a methodological limitation.
  • The PRISMA flowchart should include the reasons for exclusion during full-text screening and a PRISMA checklist should be included as an appendix.
  • The justifications for the CASP ratings should be reported in an appendix.
  • In some parts of the results section, percentages are reported as if they represent the entire population of intersex people, when they only represent the specific sample in the associated study. Please review the results section for this issue and rectify it.
  • In the paragraph on "Assigned Sex", it is reported that "41% were assigned male sex at birth, with 23% continuing to use the same male assignation". Is the second figure a percentage of the entire sample or a percentage of the sample assigned male at birth?
  • I felt that the second half of the paragraph on "Assigned Sex"  was more relevant to gender identity and should perhaps have been reported in the following paragraph.
  • In the discussion, there is a call for future research to be co-produced with intersex people. While I totally agree with this call, I think a more substantial argument in favour of it should be presented for those less familiar with what co-produced research is and why it is particularly relevant to this population.

Author Response

(The authors gave the same response as above.)
